

# Trypophobia as an urbanized emotion: comparative research in ethnic minority regions of China

Siqi Zhu[1], Kyoshiro Sasaki[2,3,4], Yue Jiang[5], Kun Qian[6] and Yuki Yamada[3]

[1] Graduate School of Human-Environment Studies, Kyushu University, Fukuoka, Japan
[2] Faculty of Science and Engineering, Waseda University, Tokyo, Japan
[3] Faculty of Arts and Science, Kyushu University, Fukuoka, Japan
[4] Japan Society for the Promotion of Science, Tokyo, Japan
[5] School of Ethnology and Sociology, Yunnan University, Kunming, China
[6] Institute of Decision Science for a Sustainable Society, Kyushu University, Fukuoka, Japan

## ABSTRACT

Trypophobia is a strong emotion of disgust evoked by clusters of holes or round objects (e.g., lotus seed pod). It has become increasingly popular and been studied since 2010s, mainly in the West and Japan. Considering this, trypophobia might be a modern emotion, and hence urbanization possibly plays key roles in trypophobia. To address this issue, we compared the degree of trypophobia between urban and less urban people in China. In an experiment, we asked participants about their degree of discomfort from trypophobic images. The results showed that trypophobia occurred in both groups, although the effect size was larger in urban than less urban people. Moreover, post-experimental interviews and post-hoc analyses revealed that older people in less urban area did not experience as much trypophobia. Our findings suggest that trypophobia links to urbanization and age-related properties.

## INTRODUCTION

Clusters of holes or round objects can induce strong feelings of disgust. This phenomenon is called *trypophobia* (*Abbasi, 2011*; *Cole & Wilkins, 2013*). It increasingly garnered public attention via the Internet and has then been investigated since the 2010s (*Abbasi, 2011*; *Cole & Wilkins, 2013*). Accordingly, it seems to be a relatively new emotional phenomenon. Previous studies have revealed the relationship between trypophobia and visual processing. *Cole & Wilkins (2013)* performed a spectral analysis of trypophobic images (e.g., lotus seed pod) and neutral images (e.g., golf cup) and showed that these images had high contrast energy at midrange spatial frequency in comparison to the neutral images. Their study indicated that spatial frequency information at midrange was involved with trypophobia. Another study revealed that low- and mid-spatial frequency information contributed to trypophobia (*Sasaki et al., 2017*). Based on these findings, trypophobic objects may be processed rapidly and unconsciously; indeed, there is

Corresponding authors
Siqi Zhu, knmoonless@gmail.com
Yuki Yamada, yamadayuk@gmail.com

empirical evidence supporting this (*Sasaki, Watanabe & Yamada, 2018*; *Shirai & Ogawa, 2019*). Taken together, early visual processing plays key roles in trypophobia.

The question remains, however: does early visual processing alone contribute to trypophobia? Several studies have discussed cognitive factors of trypophobia. Recent studies have developed the Trypophobia Questionnaire (TQ) and investigated the relationship between trypophobia and personal traits (*Chaya et al., 2016*; *Le, Cole & Wilkins, 2015*; *Imaizumi et al., 2016a*). In particular, *Imaizumi et al. (2016b)* showed that core disgust (i.e., emotional avoidance of pathogen infection) sensitivity and positively predicted the TQ score. This finding indicates that disgust contributes to trypophobia, and other findings also support this idea (*Kupfer & Le, 2018*; *Vlok-Barnard & Stein, 2017*). Based on these studies, *Yamada & Sasaki (2017)* proposed the "involuntary protection against dermatosis" (IPAD) hypothesis. According to this hypothesis, trypophobic objects evoke unpleasant emotions because their appearance is associated with dermatosis and, as a result, an avoidance reaction to pathogens is induced. Indeed, Yamada and Sasaki also provided evidence supporting the IPAD hypothesis; the history of skin problems involves trypophobic discomfort. Thus, disgust toward infectious pathogens should involve trypophobia.

Studies on trypophobia have rapidly increased, as we mentioned above. However, these studies were mainly conducted in the West and in Japan. Additionally, numerous cross-cultural studies on emotions show cultural differences in emotional processing (*Grossmann, Ellsworth & Hong, 2012*; *Grossmann et al., 2014*; *Hot et al., 2006*; *Kitayama, Mesquita & Karasawa, 2006*; *Masuda et al., 2008*; *Tanaka et al., 2010*), although these studies mainly addressed general positive and negative emotions. Recently, given technological advances, it has become possible for behavioral data to be collected from less-urbanized areas, which was previously difficult (*Takahashi, Oishi & Shimada, 2017*). Considering that trypophobia has become popular and been studied since about 2010, mainly in Western and Japanese culture areas (*Abbasi, 2011*; *Cole & Wilkins, 2013*), it might be a modern emotion that possibly involves urbanization, and people in less-urbanized areas might not experience trypophobia. To address this issue, we examined whether trypophobia occurs in less-urbanized areas in China.

In the current study, we focused on two regions in Southwest China, Yunnan Province and Guangxi Zhuang Autonomous Region. Yunnan province has mountains at its southwestern and southern part, including the Ailao and Wuliang Mountains along the southern coast of the Honghe River. In these mountainous areas, several ethnic minorities live a less-urbanized life. For example, Hani people, a mountainous ethnic group in the frontier of the country, are distributed along the Ailao and Wuliang Mountains. Dai people also live in the Honghe region, but are more concentrated in valleys or the relatively flat regions of the Honghe River basin. Guangxi Zhuang Autonomous Region is located at the southeast edge of the Yunnan–Guizhou Plateau. Yao people live in rural areas of the northwest and northeast mountainous and hilly areas. Thus, the present study examined whether people in the less urbanized areas of the Honghe region of Yunnan (i.e., Hani and Dai people) and Qibainong mountainous area in northwest of Guangxi (i.e., Yao people) experience trypophobia. As the control group, we also asked Chinese
people in an urbanized area in the Yunnan Province, China and in Fukuoka City, Japan, if they experienced trypophobia. We hypothesized that trypophobia would be more salient as urbanization progressed. Therefore, we predicted that less urban people would experience weaker discomfort from the trypophobic images than would urban people and that, in less urban people, there would be no or small difference in the degree of discomfort between the neutral and trypophobic images.

## METHODS

### Ethics statement

The present study received approval from the psychological research ethics committee of the Faculty of Human-Environment Studies at Kyushu University (approval number: 2018-002). The experiment was conducted according to the guidelines laid down in the Helsinki declaration. Informed consent was verbally obtained from all participants because some were pre-literate. Participants had the right to withdraw from the experiment at any time without providing any reason.

### Participants

We performed a two-way mixed-design analysis of variance (ANOVA) with the image type (trypophobic and neutral) as a within-participant factor and the participant group (less urban and urban people) as a between-participant factor. The required sample sizes were calculated using G*power (*Faul et al., 2007*). We mainly intended to test whether there is an interaction effect between the image type and the participant group. Thus, we performed a preliminary test to detect the interaction effect ($\alpha = 0.05$, $1-\beta = 0.80$, $f = 0.25$), and then estimated the sample size to be 34 (i.e., 17 participants per exposure condition). However, a previous study suggests that at least 20 participants are required per group to avoid Type I errors (*Simmons, Nelson & Simonsohn, 2011*). Thus, we set the minimal sample size to 40 participants (i.e., 20 per exposure condition) for the statistical analysis.

Thirty-four ethnic minorities living in less-urbanized areas (21 males and 13 females, mean age ± SEM = 48.9 ± 3.34 years) and 34 students in urban areas (11 males and 23 females, mean age ± SEM = 23.6 ± 0.36 years) participated in the experiment[1]. The ethnic minorities of the less urban group consisted of 10 Hani, 10 Dai and 14 Yao people. Dai participants' data were collected from the Qimaba town of the Huayao Dai minorities, while those of Hani participants were collected from Hani villages around Lüchun County. Both fields were in Honghe Hani and Yi Autonomous Prefecture of Yunnan Province, China. We also collected data of Yao people from the Qibainong Area of the Yao villages in Guangxi Zhuang Autonomous Region, China. The data of urban students were collected from the campus of Yunnan University in Kunming, China and Kyushu University in Fukuoka, Japan. This urban group consisted of Han (the ethnic majority in China), Yi, Jingpo and Zhuang peoples. Though the last three are relative minority groups among Yunnan and Kyushu University students, they are sufficient to be regarded as a control group because they had lived in a city for a long time and were

[1] Initially, 20 ethnic minorities living in less-urbanized areas and 22 students from Yunnan University (i.e., urban areas) participated in the experiment based on the power analysis. We obtained similar results to that of the main text. According to the reviewer's suggestion, we added additional data from 14 less urban people from Guangxi and 12 urban people from Kyushu University.

urbanized to a large extent. All of them were naive to the purpose of this experiment and reported having normal vision.

## Apparatus and stimuli

The stimuli were presented on a laptop (Dell Inspiron 7460). The resolution was $1{,}920 \times 1{,}080$ pixels, and the refresh rate was 100 Hz. The presentation of the stimuli and the collection of data were controlled by a computer. The stimuli were generated using PsychoPy3 (*Peirce, 2007*) and included 20 trypophobic and 20 neutral images ($512 \times 512$ pixels) used in previous studies (*Le, Cole & Wilkins, 2015*; *Sasaki et al., 2017*; *Yamada & Sasaki, 2017*).

## Procedure

The participants initiated each trial by pressing the spacebar on a computer keyboard. After the fixation mark was presented for 500 ms, the image stimulus appeared. They had to erase the stimulus by pressing a key when they thought that they had observed the stimulus enough. The rating scales were then presented. The participants were asked to evaluate the degree of discomfort for each image on a nine-point scale that ranged from 1 (strong discomfort) to 9 (strong comfort). Each participant performed 40 trials: two image types (trypophobic and neutral) × 20 images. Trials were randomized for each participant. Afterward, a post-experimental interview was performed. The interview began with the question of whether participants felt uncomfortable when they saw images of a single hole or clusters of holes. If they answered "Yes," the subsequent question was about the part that made them feel uncomfortable. If they answered "No," the subsequent question was about the feeling they actually experienced when they looked at these pictures and why they felt it.

## Analysis

We calculated the average rating scores of the trypophobic and neutral images for each participant. We performed a two-way mixed ANOVA on the rating score with the image type as a within-participant factor and the participant group as a between-participant factor. When the interaction between the image type and participant group was significant, we performed a test of the simple main effects. We set the significance level at $\alpha = 0.05$ and reported $\eta_p^2$ s.

## RESULTS

The results are shown in Fig. 1. The ANOVA revealed that the main effects of the image type and the participant group were significant (image type: $F_{(1, 66)} = 106.31$, $p < 0.001$, $\eta_p^2 = 0.62$; participant group: $F_{(1, 66)} = 8.92$, $p = 0.004$, $\eta_p^2 = 0.12$). Moreover, the interaction was significant ($F_{(1, 66)} = 31.04$, $p < 0.001$, $\eta_p^2 = 0.32$). The simple main effect of the image type was significant in both the participant groups (less urban people: $F_{(1, 33)} = 16.05$, $p < 0.001$, $\eta_p^2 = 0.33$; urban people: $F_{(1, 33)} = 96.99$, $p < 0.001$, $\eta_p^2 = 0.75$). Furthermore, the simple main effect of the participant group was significant for the trypophobic image ($F_{(1, 66)} = 19.29$, $p < 0.001$, $\eta_p^2 = 0.23$) while it was not significant in the neutral image ($F_{(1, 66)} = 0.24$, $p = 0.63$, $\eta_p^2 = 0.004$)[2].

[2] According to Reviewer 1's suggestion, we also performed a two-way mixed ANOVA on the rating score with the image type as a within-participant factor and the participant's location (Honghe, Kunming, Qibainong, Qujing, and Fukuoka) as a between-participant factor to test whether the variance due to the participant group (urban or less urban) was greater than that due to the participant's location. As a result, both of the main effects and interaction were significant (image type: $F_{(1, 63)} = 59.83$, $p < .001$, $\eta_p^2 = .49$; participant's location: $F_{(4, 63)} = 3.78$, $p = .008$, $\eta_p^2 = .19$; interaction: $F_{(1, 63)} = 8.08$, $p = .004$, $\eta_p^2 = .34$). The results suggest that the variance due to the participant group ($F = 8.92$) was greater than that due to the participant's location ($F = 3.78$).

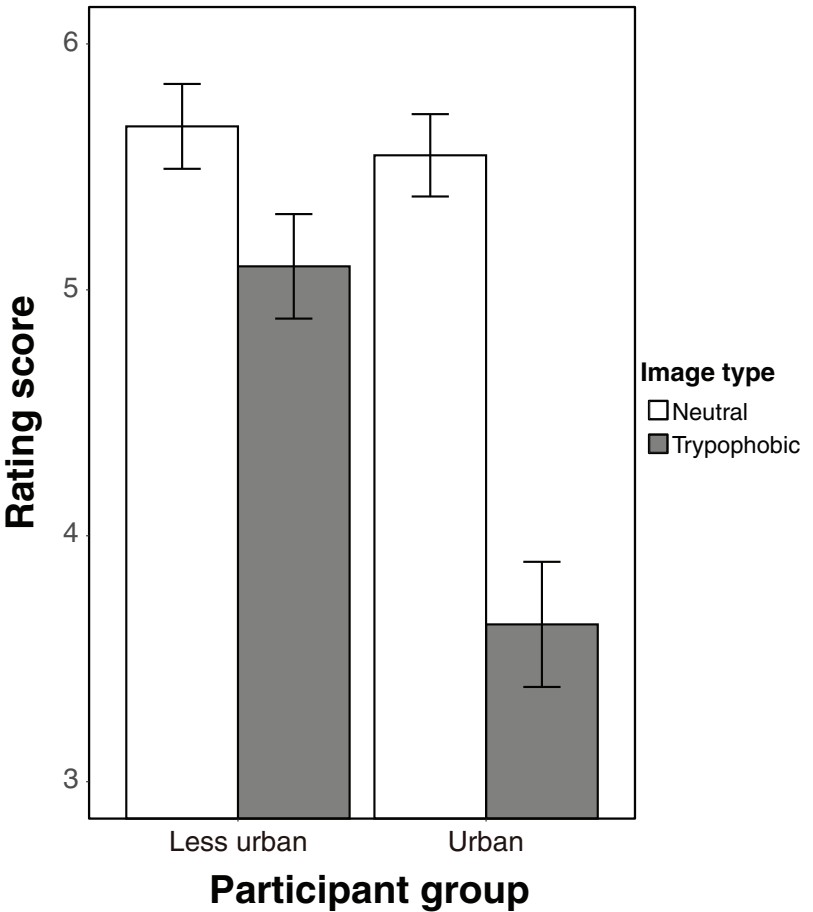

**Figure 1 The results of the experiment.** The error bars indicate standard errors of the mean.

After the experiment, participants verbally reported their impressions. Except for simple answers such as "yes" or "uncomfortable" to the questions about whether participants felt uncomfortable, a part of the other detailed reports are shown in Table 1. The full interview reports are available at the data repository (https://osf.io/wvu8z/).

## DISCUSSION

The present study demonstrated that trypophobia occurs in Chinese ethnic minorities in less urban as well as urbanized people. As mentioned above, we hypothesized that urbanization is one of the factors contributing to trypophobia, and to examine this hypothesis, a comparison of the effect sizes of trypophobia on less urban and urban people would be informative. Based on the results for each group, a smaller effect size was found for less urban ($\eta_p^2 = 0.33$) than for urban people ($\eta_p^2 = 0.75$). Briefly, trypophobia was salient in urban people in comparison with less urban people. Additionally, the rating score of the trypophobic image was significantly lower in urban than in less urban people, while there was no significant difference in the neutral image scores. Thus, the urban people experienced more discomfort from the trypophobic, but not neutral, images than did the less urban people. These results suggest that trypophobia comes from urbanization.

**Table 1 Post-experimental interviews.**

| Area | Ethnicity | Age | Gender | Report |
|------|-----------|-----|--------|--------|
| Less urban | Dai | 10 | Female | The porous images made me feel dizzy. |
| | | 41 | Female | I felt that many holes were dirty. |
| | | 40 | Female | I felt a little nauseous and scared of the porous images. |
| | | 64 | Female | No matter which image I saw, I felt good and did not feel dizzy or uncomfortable. |
| | | 72 | Male | I did not feel uncomfortable. I evaluated the impression of the images based on whether things inside the images were practically useful or whether they were ornamentally valuable. |
| | | 57 | Male | It did not matter how many holes the images contained. If the brightness of the image was dark, I felt uncomfortable. On the other hand, its brightness was high, which made me comfortable. |
| | Hani | 42 | Female | The images did not induce discomfort regardless of whether these contained many holes. I prefer plaid clothes. |
| | | 79 | Male | I did not feel uncomfortable. I actually felt that the porous images were better-looking. |
| | | 30 | Female | The image containing high-dense holes made me dizzy and thus I did not want to look at them. |
| | | 55 | Male | The images containing fewer holes were good. |
| Urban | Yi | 25 | Female | I was very disgusted. |
| | | 20 | Male | I felt nothing special about the images. |
| | Jingpo | 25 | Female | I hated the clusters of small holes, although the holes on the food were good. |
| | Han | 21 | Female | I did not feel good. |
| | | 18 | Male | The porous images were eye-catching, while I felt nothing special about the other images. |
| | | 23 | Female | When I saw the porous image, I felt as if my scalp was numb and my hair stood up. |
| | | 26 | Male | The holes would be disgusting if they were randomly arranged and thus looked worm-eaten and like acne. However, if the holes were regularly aligned like beehives, they were good. |
| | | 22 | Female | It did not matter if the porous images contained artificial objects. However, they were disgusting if they contained too many holes. |
| | | 23 | Male | Holes on natural things made me nauseous. In particular, when objects in the holes were similar to acne, I felt disgust. The dirty images also made me nauseous. |
| | | 23 | Male | When lots of holes were regularly aligned, they were disgusting. |
| | | 24 | Female | I felt nothing special about the images. When I saw the mushroom, I even wanted to eat it. |

Does urbanization alone regulate trypophobic responses? This may not be true. There are several differences in attributes and living environments between less urban and urban people; one possibility is a difference in age. In actuality, the participants' age was significantly higher in the less urban than in the urban people (Welch's $t$-test: $t(33.75) = 7.54$, $p < 0.001$, Cohen's $d = 1.83$). Did the participants' age also link to trypophobia? According to the post-experimental interview, the older less urban people do not experience much trypophobia. Moreover, the post-hoc tests showed that the differences in the rating scores (i.e., subtracting the scores of the trypophobic image from those of the neutral ones) significantly correlated with the participants' age ($r = -0.37$, $p = 0.002$), indicating that trypophobia weakened with participants' higher age. A similar (but considerably weak) pattern was found in a previous study (*Imaizumi et al., 2016a*), which used the TQ with Japanese people. Taken together, although we could not clearly conclude which factor contingent to urbanization involves trypophobia, age-related properties are possibly crucial ones.
The most interesting result of the present study was that, as mentioned earlier, even people living in less-urbanized regions strongly experienced trypophobia, suggesting its high prevalence. Here, one question arises: since when did this phenomenon exist? The study of trypophobia has been rapidly increasing since 2000. To the best of our knowledge, the oldest case of a fear of a hole, which seems to be very similar to trypophobia, was described in 1998 (*Rufo, 1998*). Our urbanization hypothesis can help to answer this question: trypophobia is a phenomenon that became stronger with urbanization, so it is likely that it was not strong enough to be discovered until global urbanization fully progressed. Furthermore, the rapid spread of trypophobia may be related to the development of image processing and information communication technologies (e.g., the Internet): someone experienced trypophobia, then posted this experience on a website (e.g., blog or forum), and as a result, trypophobia spread over the world[3]. These occurrences played a critical role in the creation, processing and dissemination of trypophobic images. In addition, because of the digital divide (*Norris, 2001*), the development produced a gap in familiarity with information technology between regions and generations, which supports our urbanized-trypophobia hypothesis: if there is any positive correlation between age and trypophobia intensity, this may be explained by the digital divide.

Because of differences in hygienic conditions, the severity of infectious diseases is significantly higher in less-urban regions than in urban areas (*Paddock, 2014*; *World Health Organization, 2002*). Are these difference in the severity of infectious diseases associated with the present findings? Based on the IPAD hypothesis (*Yamada & Sasaki, 2017*), trypophobia is assumed to be an avoidance reaction to infectious pathogens because the appearance of trypophobic images is associated with skin diseases. This theory would predict that people in less-urban areas would experience trypophobia more strongly than would people in urban areas; however, this was not the case. Therefore, the difference in the severity of infectious diseases cannot simply account for the difference in trypophobia between urban and less-urban areas. The present findings give rise to a new possibility: infection habituation. People in less-urban areas are frequently in contact with infectious pathogens (*Paddock, 2014*; *World Health Organization, 2002*) and as a result, are habituated to them. Such habituation might reduce the fear of infectious diseases and, in turn, weaken the avoidance response to apparent sources of infection, thereby attenuating trypophobia. Further experimentation on this point is thus warranted.

The present study has some limitations worth discussing. First, our experiment was conducted using a laptop computer, but the middle-aged and older people in less-urban area were embarrassed about their performances using such digital experimental device. The difficulty in operation possibly caused increased variance in the data. Moreover, the embarrassment may have somewhat contaminated and biased the emotional responses of comfort in the minority sample, although the neutral condition did not differ between the groups. Second, less-urban participants included those who did not understand Mandarin Chinese, and hence, the experimental instruction sometimes required translation by experimentally naive local guides who used local ethnic languages. This translation may have changed nuances, affecting the results. In future research, these

[3] Reviewer 2 also experienced a similar case.

problems may be solved through non-digital methods such as paper experiments and by inviting local psychological scientists to conduct experiments. Third, there were ethnic differences between urbanized and less-urbanized areas and these differences might have potential effects on our results that have not yet been discovered. Although there is no reasonable hypothesis on the differences in trypophobia among ethnic groups at this time and thus it is less likely that the ethnic differences contaminated our results, controlling these differences might be desirable in future studies.

Fourth, all participants reported that their visual acuity was normal, but their visual function was not actually measured and could not be confirmed. This may also affect the evaluation results. Indeed, provocative visual patterns in the modern urban environments possibly involve visual stress (*Wilkins, Penacchio & Leonards, 2018*), which is linked to trypophobia (*Imaizumi et al., 2016b*). Moreover, as *Sasaki et al. (2017)* revealed, trypophobia is a phenomenon that is dependent on spatial frequency. According to their study, high-frequency components only minorly contributed to the strength of trypophobia. Differences in visual function have often been argued about in studies on cultural differences in visual perception (*Ahluwalia, 1978*; *Berry et al., 2012*; *Jahoda, 1971*). If less-urban participants who participated in this study had visual characteristics biased toward processing of high-frequency components, it is possible that trypophobia as seen in the present results is underestimated.

Last but not least, through this research in the Honghe and Qibainong Areas, we found that the development of trypophobia in less-urban areas, which had been experimented with previously, greatly exceeded the initial expectations. These areas can still have some unique cultural symbols of ethnic minorities, such as residents wearing traditional ethnic costumes and holding traditional ceremonies during festivals. However, in recent times, the wooden buildings have been transformed into brick buildings and most residents skillfully use mobile phones and electronic payments. These phenomena show that urbanization and modernization are accelerating even in the minority areas along the border. Therefore, the boundaries between urban and less-urban areas in China are gradually dissolving. Nevertheless, the results of the present study also seem to tell us that even in the course of such transformations, the regional differences in terms of trypophobia still exist. Thus, testing trypophobia in some more primitive ethnic settlements, such as with the Dulong in Nujiang, Yunnan, or ethnic minority areas in Thailand, Laos and Myanmar, concerning the difference between these peripheral regions and urban areas is worth exploring in the future.

## CONCLUSIONS

The present study aimed at testing whether trypophobia occurs in people in rural areas, especially in Chinese ethnic minorities. Our hypothesis was that people in rural areas experienced weaker trypophobia than people in urban areas. Consistent with this hypothesis, weaker trypophobia occurred in rural people, suggesting that trypophobia is a relatively new emotion that has emerged through urbanization. Future research using more peoples secluded from modern civilizations is needed.

### Funding

This research was supported by JSPS KAKENHI (15H05709, 16H01866, 17H00875, 17H06342, 17J05236, 18H04199, 18K12015 and 19K14482). The funders had no role in study design, data collection and analysis, decision to publish, or preparation of the manuscript.

### Grant Disclosures

The following grant information was disclosed by the authors:
JSPS KAKENHI: 15H05709, 16H01866, 17H00875, 17H06342, 17J05236, 18H04199, 18K12015 and 19K14482.

### Competing Interests

The authors declare that they have no competing interests.

### Author Contributions

- Siqi Zhu conceived and designed the experiments, performed the experiments, analyzed the data, prepared figures and/or tables, authored or reviewed drafts of the paper, and approved the final draft.
- Kyoshiro Sasaki conceived and designed the experiments, analyzed the data, prepared figures and/or tables, authored or reviewed drafts of the paper, and approved the final draft.
- Yue Jiang performed the experiments, authored or reviewed drafts of the paper, and approved the final draft.
- Kun Qian conceived and designed the experiments, analyzed the data, prepared figures and/or tables, authored or reviewed drafts of the paper, and approved the final draft.
- Yuki Yamada conceived and designed the experiments, analyzed the data, prepared figures and/or tables, authored or reviewed drafts of the paper, and approved the final draft.

### Human Ethics

The following information was supplied relating to ethical approvals (i.e., approving body and any reference numbers):

The present study received approval from the psychological research ethics committee of the Faculty of Human-Environment Studies at Kyushu University (approval number: 2018-002).

### Data Availability

The datasets are available at Open Science Foundation: Siqi Zhu, Kyoshiro Sasaki, Yue Jiang, Kun Qian and Yuki Yamada. 2020. "Trypo and Tribe." OSF. February 18. DOI 10.17605/OSF.IO/WVU8Z.

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
