# Peer review of "Trypophobia as an urbanized emotion: comparative research in ethnic minority regions of China"

_PeerJ, doi:10.7717/peerj.8837_

## Round 0.1 · original submission · Major Revisions

Dear Authors,

Please perform the relevant revisions to the manuscript with the comments of the peer reviewers.

Thanking You.

Reviewer 1 ·

Basic reporting

Trypophobia is not simply due to the internet because many of the individuals with trypophobia I have talked to experienced trypophobia long before the internet was born. The internet has, however, been responsible for bringing isolated sufferers together.

Experimental design

Needs further work.
The report addresses a sensible question and the results are interesting. The statistical power in the studies is low, however. I would like to see a larger sample of individuals, and more than one sample of locations in order to be sure of the estimates of the prevalence of trypophobia in city versus country dwellers. One needs to be able to estimate the within-category variability.

Validity of the findings

See below

Additional comments

Provocative visual patterns are prevalent in the modern urban environment, see Wilkins, A.J. Penacchio, O. and Leonards, U. (2018). The built environment and its patterns: a view from the vision sciences. SDAR Journal of Sustainable Design and Applied Research, 6, (1) 41-48. These patterns tend not to occur in nature. They can induce visual distortions, headaches and seizures, a phenomenon known as visual stress. Visual stress is a component of trypophobia, at least in so far as a correlation between visual discomfort and trypophobia has been found, see Imaizumi, S., Furuno, M., Hibino, H., & Koyama, S. (2016). Trypophobia is predicted by disgust sensitivity, empathic traits, and visual discomfort. SpringerPlus, 5(1), 1449. doi: 10.1186/s40064-016-3149-6 So one possible argument the authors might wish to make is that the modern urban environment has increased the levels of visual stress generally, and that trypophobia is one result of this increase.

·

Basic reporting

Not professional English.

Good background.

Results fine.

Experimental design

All fine.

Validity of the findings

All fine.

Additional comments

In this paper the authors report results from a single study in which sensitivity to trypophobia stimuli was examined in people who live in an urban environment and those living in a less-urban environment. Participants were shown a series of trypophobic images (and controls) and asked to rate each for discomfort. The central results showed that both populations showed trypophobia but the size of this effect was greater in the urbanised population. No such difference occurred for the control images. The authors thus conclude that urbanisation plays a contributing role in trypophobia.

Evaluation.

I like the work and do think it will make a nice contribution to the journal. I only have a few points.

1) I don’t think there is a confound in their experiment but the authors constantly suggest that such a confound exists. They state that they are comparing an urbanised population with ethnic minorities (then conclude that urbanisation is important), but this makes it sound like we can’t know whether the effect they observe is due to urbanisation or how prevalent the ethnic group is. In other words, it should be either ethnic minority versus ethnic majority or urban versus non-urban, not ethnic minority versus urban.

2) The authors give a very weak explanation, concerning clothes, as to why older people showed less trypophobia. The reason that older people are less trypophobic is simply that many phobias are known to reduce as age increases (e.g., Fredrikson, et al. 1996). I would definitely drop the clothes theory, which is also too anecdotal.

3) The authors also give a very weak explanation as to why interest and knowledge in trypophobia only occurred in the past few years, including ‘bad luck’, ‘researchers negligence’ and an urbanisation-based account. These are all a bit bizarre and need to be dropped (definitely the ‘negligence’ one; perhaps lost in translation but ‘negligence’ is when someone does something very seriously bad). The truth is that the internet has helped to publicise it; it’s a good meme. Also, when I gave the first presentation on the phenomenon, in 2012, a journalist from the Washington Post was present and wanted to run a story on it. This then helped to publicise the paper we published in Psych Science. Its as simple as that.

4) The English needs tidying up. E.g., Line 42. “This phenomenon is called as trypophobia”

5) Line 74. The authors state that “ there are numerous cross-cultural studies on emotions showing cultural difference in emotional processing”. I think a couple of sentences are needed here stating which emotions are particularly different.

---

## Round 0.2 · Major Revisions

Please do the needed revisions as per the peer reviewers' comments. In particular, please address the sample size issues raised by Reviewer 1.

Reviewer 1 ·

Basic reporting

This is a second review

Experimental design

The authors appear to have misunderstood my objection to their sample size and distribution, which has nothing to do with power analysis, p-hacking or other considerations. In order to make a claim that the differences between samples are due to their being "urban" or "rural" one needs more than one sample of individuals from each category, particularly when the individuals are from single locations. These locations may simply not represent the category assigned to them.

Validity of the findings

This is a second review

Additional comments

Personally, I do not feel that the paper is ready for publication. The sample size is small and the samples are not necessarily representative. More work needs to be done.

·

Basic reporting

See below

Experimental design

See below

Validity of the findings

See below

Additional comments

This is a revision to a ms I reviewed previously which I was largely happy with. The authors have made a good job of the revision. I recommend publication as is.

---

## Author Rebuttal · Round 0.2

16, December 2019

Dr. Jafri Abdullah
Academic Editor
Dear Dr. Abdullah:

We have read the reviewers' comments regarding our manuscript and we greatly appreciate the valuable feedback. Please find attached the revised manuscript, #42574, entitled "Trypophobia as an urbanized emotion: Comparative research in ethnic minority regions of China," by Zhu et al. Based on the reviewers' comments, the manuscript has been substantially revised. Below, please find our responses to the individual reviewer comments.

**Responses to Reviewer 1**

**#**     **Comments & Replies**

1-1     **_Basic reporting_**

*Trypophobia is not simply due to the internet because many of the individuals with trypophobia I have talked to experienced trypophobia long before the internet was born. The internet has, however, been responsible for bringing isolated sufferers together.*

**Reply:** We cordially thank Reviewer 1 for reviewing our manuscript and providing these important comments. In our manuscript, we did not intend to deny the possibility that trypophobia occurred before the Internet was born. Actually, in the original manuscript we did introduce an observational report (Rufo, 1998) that was related to trypophobia and published before the Internet spread globally. Just as Reviewer 1 notes, we think that the Internet contributed to the *familiarization* of trypophobia.

1-2     **_Experimental design_**

*Needs further work.*

*The report addresses a sensible question and the results are interesting. The statistical power in the studies is low, however. I would like to see a larger sample of individuals, and more than one sample of locations in order to be sure of the estimates of the prevalence of trypophobia in city versus country dwellers. One needs to be able to estimate the within-category variability.*

**Reply:** We appreciate the important comment. However, adding data here, at least

without statistically valid grounds, causes many problems, for the following reason.

1. **It makes power analysis meaningless**. The statistical power of this study is clearly sufficient. We performed a power analysis before starting the experiment. The settings we established were quite common ($\alpha$ = .05, 1-$\beta$ = .80, $f$ = .25). In addition, in order to obtain a higher power, a sample larger than the calculated required sample size ($n$ = 34) was actually collected ($n$ = 42). This sample size has a fairly high power (1-$\beta$ = 0.89). Moreover, as a result of the analysis using the present data, it was shown that the power is maximal (1-$\beta$ = 1) and hence, the probability of making a type II error is extremely low. Despite these statistical bases, determining a new sample size based on intuition and impression is a practice of making the prior power analysis meaningless and denying the sample size design itself.

2. **It is p-hacking**. Increasing the sample size to increase the statistical power *after* the experiment has been conducted is a questionable research practice called *p*-hacking, which we do not want to engage in.

3. **It may include HARKing**. We did not develop any hypotheses about ethnic diversity or cultural differences. In addition, no one has suggested any alternative reasonable hypothesis on such ethnic diversity (rather, Reviewer B noted, "I don't think there is a confound in their experiment"). Nevertheless, increasing the sample size "to estimate the within-category variability" is not justified unless our original hypothesis has changed. However, we do not want to change our original hypothesis due to personal intuition. Moreover, changing our hypothesis *after* the experiment is a questionable research practice called HARKing (Hypothesizing after the Results are Known), which we do not want to engage in.

For these reasons, we did not add any data to the current study. However, we understand that descriptions in the manuscript could have triggered the present question. For example, there may have been a description that led to the impression that we had a hypothesis about some cultural differences between ethnic groups. We carefully reviewed the manuscript and made overall corrections to minimize such misunderstandings. We would like to express our sincere gratitude to the reviewer for alerting us to this point.

1-3     ***Comments for the Author***

*Provocative visual patterns are prevalent in the modern urban environment, see Wilkins, A.J. Penacchio, O. and Leonards, U. (2018). The built environment and its patterns: a view from the vision sciences. SDAR Journal of Sustainable Design and Applied Research, 6, (1) 41-48. These patterns tend not to occur in nature. They can*

*induce visual distortions, headaches and seizures, a phenomenon known as visual stress. Visual stress is a component of trypophobia, at least in so far as a correlation between visual discomfort and trypophobia has been found, see Imaizumi, S., Furuno, M., Hibino, H., & Koyama, S. (2016). Trypophobia is predicted by disgust sensitivity, empathic traits, and visual discomfort. SpringerPlus, 5(1), 1449. doi: 10.1186/s40064-016-3149-6 So one possible argument the authors might wish to make is that the modern urban environment has increased the levels of visual stress generally, and that trypophobia is one result of this increase.*

**Reply:** As Reviewer 1 suggested, visual stress stemming from modern urban environments might be involved with trypophobia. Based on this comment, we discuss this issue in the revised manuscript (lines: 267–269).

**Responses to Dr. Geoff Cole (Reviewer 2)**

| **#** | **Comments & Replies** |
|---|---|
| 2-1 | *In this paper the authors report results from a single study in which sensitivity to trypophobia stimuli was examined in people who live in an urban environment and those living in a less-urban environment. Participants were shown a series of trypophobic images (and controls) and asked to rate each for discomfort. The central results showed that both populations showed trypophobia but the size of this effect was greater in the urbanised population. No such difference occurred for the control images. The authors thus conclude that urbanisation plays a contributing role in trypophobia.* |

*Evaluation.*

*I like the work and do think it will make a nice contribution to the journal. I only have a few points.*

**Reply:** We would like to thank Reviewer 2 for appropriately understanding our manuscript and for providing positive feedback. We are happy to address the points kindly raised by the reviewer to improve the manuscript.

| | |
|---|---|
| 2-2 | *I don't think there is a confound in their experiment but the authors constantly suggest that such a confound exists. They state that they are comparing an urbanised population with ethnic minorities (then conclude that urbanisation is important), but this makes it sound like we can't know whether the effect they observe is due to urbanisation or how prevalent the ethnic group is. In other words, it should be either* |

*ethnic minority versus ethnic majority or urban versus non-urban, not ethnic minority versus urban.*

**Reply:** Although our main aim was to examine whether the degree of trypophobia would be different between urban and less-urban areas, some descriptions might have caused the misunderstanding that we were trying to address the differences in trypophobia among ethnic groups. We have revised these misleading descriptions. Actually, as Reviewer 2 mentioned, we also think that there was no confound in our experiment. In the first place, there is no reasonable hypothesis on the differences in trypophobia among ethnic groups at this time. However, we cannot deny potential effects related to ethnic differences that have not yet been hypothesized; thus, we discuss these effects in the Discussion in the revised manuscript (lines 259–264). We appreciate this valuable comment.

2-3    *The authors give a very weak explanation, concerning clothes, as to why older people showed less trypophobia. The reason that older people are less trypophobic is simply that many phobias are known to reduce as age increases (e.g., Fredrikson, et al. 1996). I would definitely drop the clothes theory, which is also too anecdotal.*
       **Reply:** We agree with this comment and have deleted this part from the revised manuscript.

2-4    *The authors also give a very weak explanation as to why interest and knowledge in trypophobia only occurred in the past few years, including 'bad luck', 'researchers negligence' and an urbanisation-based account. These are all a bit bizarre and need to be dropped (definitely the 'negligence' one; perhaps lost in translation but 'negligence' is when someone does something very seriously bad). The truth is that the internet has helped to publicise it; it's a good meme. Also, when I gave the first presentation on the phenomenon, in 2012, a journalist from the Washington Post was present and wanted to run a story on it. This then helped to publicise the paper we published in Psych Science. Its as simple as that.*
       **Reply:** As Reviewer 2 pointed out, these explanations of "bad luck" and "researchers' negligence" seem to be weak and the expressions were misleading. We have deleted these descriptions from the revised manuscript. As Reviewer 2 mentioned, the Internet contributes to the publicization of trypophobia. That is, people might have locally felt and experienced trypophobia before the Internet, and the Internet possibly created bridges between such people. We added descriptions about these issues in the Discussion (lines 221–225).

2-5    *The English needs tidying up. E.g., Line 42. "This phenomenon is called as*

*trypophobia"*

**Reply:** We have modified this text as follows: *This phenomenon is called trypophobia*. Moreover, a native English speaker and professional editor has once again checked our manuscript.

2-6     *Line 74. The authors state that " there are numerous cross-cultural studies on emotions showing cultural difference in emotional processing". I think a couple of sentences are needed here stating which emotions are particularly different.*

**Reply:** Thank you for this comment. These previous studies mainly addressed general positive and negative emotions among various areas and thus it is difficult to state which emotions are particularly different overall. We have added the following statement: *Although these studies mainly addressed general positive and negative emotions*.

Again, we would like to express our gratitude to the reviewers for all of their thoughtful and constructive comments. We hope that our revised manuscript is now suitable for publication in *PeerJ*.

Sincerely,

Siqi Zhu and Yuki Yamada
Kyushu University
744 Motooka, Nishi-ku, Fukuoka 819-0395, Japan
Phone/Fax number: +81-92-802-5837
E-mails: knmoonless@gmail.com (S.Z.) and yamadayuk@gmail.com (Y.Y.)

---

## Round 0.3 · Minor Revisions

Dear Authors,

Please revise the manuscript as per the suggestions from Reviewer 1.

Thanking You.

Reviewer 1 ·

Basic reporting

see below

Experimental design

see below

Validity of the findings

see below

Additional comments

I am delighted that the authors have collected additional data so that they can separate the categories of urban and rural from the particular locations they used. This has greatly improved the paper, and it is now nearly publishable. I would however, like to suggest one additional analysis. I think there are now several groups, depending on location. It would be important to analyse the data using an analysis of variance that grouped the individuals by the several locations, so that the reader can appreciate that the categories of urban and rural transcend the locations from which they are derived. This will involve small changes to the analyses they have undertaken and the measures of error involved. Is the variance due to category (urban/rural) greater than the variance due to location?

---

## Round 0.4 · accepted · Accept

Congratulations! The manuscript has been accepted. Thank you.

Reviewer 1 ·

Basic reporting

OK

Experimental design

OK

Validity of the findings

Improved

Additional comments

The authors have greatly improved their paper and it will now make a valuable contribution to the literature.